# Correlation between Blood Flow and Temperature of the Ocular Anterior Segment in Normal Subjects

**DOI:** 10.3390/diagnostics10090695

**Published:** 2020-09-15

**Authors:** Takashi Itokawa, Takashi Suzuki, Yukinobu Okajima, Tatsuhiko Kobayashi, Hiroko Iwashita, Satoshi Gotoda, Koji Kakisu, Yuto Tei, Yuichi Hori

**Affiliations:** 1Department of Ophthalmology, Toho University Graduate School of Medicine, Tokyo 143-8541, Japan; takashi.itokawa@med.toho-u.ac.jp (T.I.); iwashita@triangle-aec.com (H.I.); yuuto.tei@med.toho-u.ac.jp (Y.T.); 2Department of Ophthalmology, School of Medicine, Toho University, Tokyo 143-8541, Japan; takashisuzuki58@gmail.com (T.S.); yukinobu.okajima@med.toho-u.ac.jp (Y.O.); tatsuhiko.kobayashi@med.toho-u.ac.jp (T.K.); satoshi.gotoda@med.toho-u.ac.jp (S.G.); kouji.kakisu@med.toho-u.ac.jp (K.K.); 3Ishizuchi Eye Clinic, Ehime 792-0811, Japan

**Keywords:** temperature, blood flow, cornea, conjunctiva, eyelid skin, warm compress, capsaicin

## Abstract

Purpose: To determine a correlation between temperature and blood flow in the ocular anterior segment, and their effects on corneal temperature. Methods: In experiment 1, we recruited 40 eyes and measured the temperature and blood flow in the ocular anterior-segment (upper/lower eyelid skin, palpebral and bulbar conjunctiva, and cornea) before and after application of warm compresses. In experiment 2, we recruited 20 eyes and measured the same tissues before and during stimulation using water and capsaicin solution in the oral cavity. Results: In experiment 1, the temperatures of the upper/lower eyelid skin and cornea increased significantly until 15 min after the application of the warm compress; the temperatures of the palpebral and bulbar conjunctiva increased significantly until 10 min. The blood flow in the upper/lower eyelid skin and bulbar conjunctiva increased significantly until 10 min, and that of the palpebral conjunctiva increased significantly until 15 min. In experiment 2, the temperatures were correlated significantly with the blood flow in the upper and lower eyelid skin and palpebral and bulbar conjunctiva. The temperature of all locations and palpebral conjunctival blood flow contributed independently to the corneal temperature. Conclusions: In the ocular anterior segment, the temperature and blood flow were correlated significantly, and contributed to the corneal temperature.

## 1. Introduction

The human body maintains homeostasis, while body temperature and blood flow adapt to changes in the external environment [1]. When disease develops, the homeostatic balance breaks down. It is well known that the pathophysiology of various systemic diseases such as gingivitis [2], arteriosclerosis [3,4], and circulatory shock [5] can be evaluated using temperature and/or blood flow. Similar to other tissues, the ocular anterior segment also adapts when it is exposed to the external environment.

The temperatures of the ocular anterior-segment tissue, i.e., cornea, conjunctiva, and eyelid skin, are useful for assessing physiological and pathological conditions. When evaluating the corneal temperature, two parameters are important: one is the corneal temperature immediately after eye opening and the other is the change in corneal temperature during the seconds after eye opening. Many researchers have reported that the changes in corneal temperature are correlated significantly with the tear film stability [6,7]. We also reported the correlation in patients who wear soft contact lens and after cataract surgery [8,9]. Numerous studies have reported that the corneal temperature varies with age [10], blinking [6,11], tear film thickness [12], and external temperature [13]. In addition, the palpebral conjunctival temperature decreases in association with meibomian gland dysfunction [14], the eyelid skin temperature decreases in dry eye [15], and the bulbar conjunctival temperature increases in allergic conjunctivitis [16]. Moreover, Parra et al. reported that the corneal temperature regulates basal tear secretion and plays a key role in maintaining the homeostasis tear film [17].

With improvements in the technology of blood flow measurement, the conjunctival blood flow has come under scrutiny [18,19,20]. Chen et al., using functional slit-lamp biomicroscopy (FSLB) that measures the movement of red blood cells, reported that dry eye increased the conjunctival blood flow due to inflammation [20]. They also reported that conjunctival blood flow velocity increased after wind stimuli despite no change in the vessel diameter [21]. One device used to measure the blood flow velocity is laser speckle flowgraphy (LSFG), which is based on the movement of red blood cells; the device evaluates the blood flow velocity by calculating the velocity of the red blood cells as the mean blur rate (MBR) of the speckle pattern [22]. The MBR measured by LSFG is a relative value, but according to previous studies, the MBR and other blood flow measurement techniques, which are expressed as absolute values, were correlated [23,24,25]. In ophthalmology, the blood flow rate has been used for various retinal diseases such as glaucoma [26], macular edema [27], diabetic retinopathy [28], and retinopathy of prematurity [29]. In addition, the relationship between systemic disease and retinal blood flow also has been studied [30]. Moreover, LSFG has been used not only in the ophthalmic field but also in relation to the feet [31], face [32], fingers [25], and gums [33]. Kashima and Hayashi reported that capsaicin stimuli in the oral cavity increased the blood flow in the facial area including the eyelid skin [32].

Other research has shown that temperature and blood flow are correlated in the forearms [34] and legs [35]. It is also important to understand the relationship between temperature and blood flow in the ocular anterior segment to understand the physiologic function and disease pathology. Duench et al. reported that diurnal variations in the bulbar conjunctival blood flow and temperature at almost the same location indicated the same tendency [19]. However, to the best of our knowledge, no study has reported the relationship between blood flow and temperature at the same location in the anterior-segment tissues and how they affect the corneal temperature, an avascular tissue. The purpose of this study was to investigate the relationship between blood flow and temperature in each location of the ocular anterior segment and their effect on corneal temperature.

## 2. Methods

### 2.1. Subjects

We enrolled 40 right eyes of 40 normal subjects (20 men, 20 women; 23.4 ± 2.8 years; range, 20–30 years) in experiment 1 and 20 right eyes of 20 normal subjects in experiment 2 (10 men, 10 women; 24.4 ± 3.2 years; range, 20–30 years). We included normal subjects who did not wear contact lenses. The exclusion criteria were a history of allergic keratoconjunctivitis, dry eye, ocular injuries, infectious keratitis, and ocular surgery. The diagnosis of dry eye was established based on a noninvasive tear breakup time (NIBUT) of less than 5 s and the Dry Eye-Related Quality of Life (DEQS) Score questionnaire with a score of more than 33 [36]. These study protocols were approved by the Ethics Committee of Faculty of Medicine, Toho University (experiment 1. #27067, 25/02/2016; experiment 2, #A18032, 15/08/2018), and the procedures used conformed to the tenets of the Declaration of Helsinki. All patients provided informed consent after they received an explanation of the possible study consequences.

### 2.2. Experiment 1

This experiment investigated the blood flow when the temperature of the anterior segment increased in response to an eyelid warming device. We used the most effective commercially available warm compress device in Japan, i.e., a warming eye pillow (Azuki no Chikara, Kiribai Chemical, Osaka, Japan) [37]. The surface temperature profiles of the warmed device (*n* = 10) microwaved for 30 s at 600 watts were evaluated for 5 min after microwaving. We used both the room temperature device (control) and warmed device on the same subject. The warmed and room temperature devices were assigned randomly and applied to each subject for 5 min on the same day. We used over the entire orbital area with the eye opening in natural blinking. The subjects rested for 20 min between the use of the two devices. We measured the temperature and blood flow in the ocular anterior segment, i.e., in the upper and lower eyelid skin, palpebral and bulbar conjunctiva, and central cornea. The heart rate (beats/minute [bpm]) and mean arterial blood pressure (MABP) (mmHg) calculated from the following formula diastolic blood pressure + (systolic blood pressure-diastolic blood pressure)/3 also were measured as systemic parameters. All examinations were completed within 2 min. The measurements were conducted before and immediately after removal and 5, 10, 15, and 20 min after removal of each device. This study was conducted between 16:00 and 18:00; the temperature (26.2 ± 1.3 °C) and humidity (24.5 ± 6.8%) were maintained at constant levels in the room in which the measurements were obtained.

### 2.3. Experiment 2

Experiment 2 investigated the temperature when the blood flow in the ocular anterior segment increased in response to oral cavity stimulation. In this study, we used pure water and capsaicin solutions as in a previous study [32], in which Kashima and Hayashi reported that capsaicin used as a stimulus in the oral cavity increased the blood flow of the entire area of facial skin including the eyelids. Capsaicin (Wako Chemical, Osaka, Japan) solution was prepared by diluting 3.3 mM capsaicin in 70% ethanol solution with water. This solution was diluted with water to achieve a 150-μM capsaicin concentration [32]. The measurements were conducted before and during oral cavity stimulation with 1 ml of water and the capsaicin solution. We measured the temperature, blood flow, and NIBUT. The heart rate (bpm) and MABP (mmHg) also were measured as systemic parameters. We also scored the pain symptoms using a visual analog scale with scores that ranged from 0 (no symptoms) to 100 (strongest symptoms). All examinations were completed within 2 min. The measurements were conducted in the following order: first during water application followed by capsaicin solution; the subjects rested for 20 min between the two solutions. This study was conducted between 16:00 and 18:00; the temperature (25.7 ± 0.6 °C) and humidity (47.7 ± 7.4%) remained constant in the room in which the measurements were obtained. 

### 2.4. Blood flow Measurement by LSFG

The blood flow in the ocular anterior segment (i.e., upper and lower eyelid skin, palpebral and bulbar conjunctiva) was measured using laser speckle flowgraphy-ocular anterior segment (LSFG-OAS, Softcare, Fukuoka, Japan). The LSFG-OAS was a modified version of the LSFG-Perfusion Function Imager (PFI) for the skin, not for the fundus, and the optics were redesigned so that the laser irradiated on the ocular anterior segment was imaged on the charge-coupled device (CCD) camera sensor. In addition, we conducted the measurements by adjusting the image angle and value of the LSFG-PFI to the ocular anterior segment. The mechanism of LSFG has been reported previously [25]. Briefly, LSFG-OAS is comprised of a probe equipped with an ordinary CCD camera (710 (width) × 480 (height) pixels) and a 830-nm diode laser. A speckle pattern forms as a result of interference with the scattered ray reflected on the red blood cells in the blood vessels of the anterior segment. The speckle pattern changes depending on the temporal changes. We evaluated the relative blood flow velocity as the MBR using the LSFG analyzer software (version 3.2.0.1, Softcare, Fukuoka, Japan). The eyelid skin was measured for 10 s with natural blinking with the eye in primary gaze (Figure 1A). To calculate the MBR, 60 pixels in a circle 80 pixels away from the upper and lower eye lid margin was set. The conjunctiva was measured for 4 s with the patient looking about 40 degrees superiorly without blinking (Figure 1B). To calculate the MBR, 60 pixels in a circle 20 pixels away from the limbus on the inferior bulbar conjunctiva and 40 pixels in a circle on the palpebral conjunctival margin were set. The MBR analyzed the points at which two or more heartbeats could be measured stably (Figure 1C,D). In experiments 1 and 2, the blood flow measurement before stimulation was measured five consecutive times. The average of the central three data points was analyzed as the MBR and the reproducibility of the MBR was assessed by determining the coefficient of variation and intraclass correlation coefficient (ICC). After stimulation, the MBR was measured once because of changes occurring over time.

### 2.5. Temperature Measurements

The temperatures of the tissues in the ocular anterior segment (i.e., upper and lower eyelid skin, palpebral and bulbar conjunctiva, and cornea) were measured using a noninvasive ocular surface thermographer (TG-1000, Tomey Corporation, Nagoya, Japan) [6,9,16]. The subjects were instructed to close their eyes for 5 s. We then measured the entire anterior segment immediately after eye opening. The measurement location was the same as that measured by the LSFG. The temperature of the ocular anterior segment was analyzed quantitatively in a 3-mm-diameter area of the palpebral conjunctiva and in a 4-mm-diameter area of the upper and lower eyelid skin, bulbar conjunctiva, and central cornea.

### 2.6. Measurement of Tear Film Stability

A tear film interferometer (DR-1α, Kowa Co. Ltd., Tokyo, Japan) with low magnification (7.2 × 8.0 mm) was used to measure the tear film stability (NIBUT) [8,38]. The subjects were asked to blink naturally and keep their eyes open for 10 s. The NIBUT was recorded once to avoid reflex tearing and defined as the time until observation of the first break in the tear film [8,9,38]. In subjects with no breaks during the 10 s observation period, the NIBUT was recorded as 10 s.

### 2.7. Statistical Analysis

The data are expressed as the mean ± standard deviation. In experiment 1, repeated measures analysis of variance was used to compare the temporal changes in the heart rate, MABP, temperature, and blood flow. Multiple comparisons were performed using Dunnett’s test when significant differences were identified. In experiment 2, the paired *t*-test was used to compare the changes in temperature and blood flow using pure water and capsaicin solutions; the correlation between the blood flow and temperature was analyzed by Pearson’s correlation coefficient. Multiple regression analysis was used to identify the independent factors associated with the corneal temperature. The value *p* < 0.05 was considered significant. To analyze the data, we used JMP version 11 statistical analysis software (SAS Institute, Inc., Cary, NC, USA).

## 3. Results

### 3.1. Experiment 1

The NIBUT and DEQS of recruited subjects were, respectively, 7.6 ± 3.0 s and 8.6 ± 12.1. The surface temperature following application of the warmed device decreased from 55.25 °C to 47.14 °C in 5 min (Figure 2).

The time-course changes in temperature and blood flow are shown in Figure 3 and Figure 4. The temperature of the upper (immediately after, 5 min, and 10 min; *p* < 0.01, 15 min, *p* < 0.05, Dunnett’s test) (Figure 3A) and lower (immediately after, 5 min, and 10 min; *p* < 0.01, 15 min, *p* < 0.05) (Figure 3B) eyelid skin and cornea (immediately after, 5 min, and 10 min, *p* < 0.01, 15 min, *p* < 0.05) (Figure 4) increased significantly until 15 min after the removal of the warmed device; and the palpebral and bulbar conjunctival temperature increased significantly until 10 min after the removal of the warmed device (immediately after (palpebral and bulbar conjunctiva; 1.40 and 1.59 °C), 5 min (0.65 and 0.73 °C) and 10 min (0.36 and 0.47 °C), *p* < 0.01 for both comparisons). Application of the room temperature device did not result in a significant increase in any part of the anterior segment.

The respective coefficients of variation of the upper and lower eyelid, palpebral conjunctiva, and bulbar conjunctiva blood flow were, respectively, 6.7 ± 4.1%, 7.2 ± 4.7%, 8.4 ± 3.9%, and 7.6 ± 4.3%. The respective ICCs were 0.87, 0.89, 0.90, and 0.92. The rates of change of the upper (immediately after, 5 and 10 min, *p* < 0.01) (Figure 3C) and lower (immediately after and 5 min, *p* < 0.01, 10 min, *p* < 0.05) (Figure 3D) eyelid skin and bulbar (immediately after (116.1%), *p* < 0.01, 5 min (110.1%) and 10 (108.8%) min, *p* < 0.05) conjunctiva increased significantly until 10 min after the removal of the warmed device, and the blood flow in the palpebral conjunctiva increased significantly until 15 min after the removal of the warmed device (immediately after (119.7%), 5 min (113.6%), 10 (116.0%) min, *p* < 0.01, and 15 (111.6%) min, *p* < 0.05). Application of the room temperature device did not result in a significant increase in any part of the anterior segment. The heart rate and MABP using the room temperature and warmed devices did not differ significantly over time (data not shown).

### 3.2. Experiment 2

The NIBUT and DEQS of the subjects, were, respectively, 7.7 ± 2.9 seconds and 6.6 ± 9.7. The changes in blood flow in the upper (water vs. capsaicin, 99.5% vs, 138%, *p* < 0.01, paired *t*-test) (Figure 5A) and lower (98.6% vs. 165.6%, *p* < 0.01) (Figure 5B) eyelid skin, palpebral (108.4% vs. 195.4%, *p* < 0.01) (Figure 5C) and bulbar (99.9% vs. 116.7%, *p* < 0.01) (Figure 5D) conjunctiva between before and during capsaicin stimulation increased significantly compared to those obtained using water stimulation.

The changes in temperature in the upper (water vs. capsaicin, 0.00 vs. 0.20 °C, *p* < 0.01) (Figure 5E) and lower (0.01 vs 0.23 °C, *p* < 0.01) (Figure 5F) eyelid skin, palpebral conjunctiva (−0.04 vs. 0.38 °C, *p* < 0.01; Figure 5G), bulbar conjunctiva (0.02 vs. 0.31 °C, *p* < 0.01) (Figure 5H), and cornea (0.08 vs. 0.30 °C, *p* < 0.01) (Figure 6) between before and during capsaicin stimulation were significantly higher than those obtained with water stimulation. A representative case of blood flow and temperature in subjects obtained before and during capsaicin stimulation are shown in Figure 7. There are clear increase of temperature and blood flow of upper/lower eyelid skin, palpebral and bulbar conjunctiva and cornea.

The pain score (water vs. capsaicin, 2.0 vs. 71.0, *p* < 0.01), heart rate (67.6 vs. 70.6, *p* < 0.05), and MABP (87.1 vs. 97.7, *p* < 0.01) during capsaicin stimulations increased significantly, while the NIBUT did not differ significantly between the two solutions.

We found a significant weak positive correlation between the changes in the blood flow and the changes in temperature before and during stimulation in the upper (r = 0.467, *p* < 0.01, Pearson’s correlation coefficients) (Figure 8A) and lower (r = 0.313, *p* < 0.05) (Figure 8B) eyelid skin, palpebral conjunctiva (r = 0.635, *p* < 0.01) (Figure 8C), and bulbar conjunctiva (r = 0.313, *p* < 0.05) (Figure 8D).

The pain score was correlated significantly with the blood flow in the upper and lower eyelid skin, palpebral conjunctiva (*p* < 0.001 for the three comparisons), and bulbar conjunctiva (*p* = 0.01) and the temperatures of those areas (*p* = 0.01, *p* < 0.05, *p* < 0.001, *p* = 0.01, respectively). The corneal temperature was correlated significantly with the blood flow in the upper eyelid skin and palpebral conjunctiva (*p* < 0.05 for both comparisons) and the temperature of the upper and lower eyelid skin, palpebral conjunctiva, and bulbar conjunctiva (*p* < 0.001 for the four comparisons) (Table 1).

Table 2 shows the results of multiple regression analyses for factors that contributed independently to the corneal temperature. Those were the temperature of the upper (β = 0.291, *p* < 0.01) and lower (β = −0.369, *p* < 0.01) eyelid skin, palpebral (β = 0.244, *p* < 0.05) and bulbar (β = 0.762, *p* < 0.001) conjunctiva, and palpebral conjunctival blood flow (β = −0.159, *p* < 0.05).

## 4. Discussion

In ocular anterior-segment diseases, the changes in the blood flow [21] and temperature [6] are useful for evaluating the disease pathology. However, the relationship between the two parameters has not been clarified. In this study, we recruited normal subjects and investigated the response of the blood flow when the temperature increased via warm compresses and the response of the temperature when the blood flow increased via oral cavity stimulation. With both approaches, there were correlations between the temperature and blood flow in the ocular anterior-segment tissues, i.e., the eyelid skin and palpebral and bulbar conjunctiva. Moreover, the bulbar conjunctival temperature had the greatest effect on the corneal temperature.

The upper and lower eyelid skin temperature increased, respectively, by 2.02 °C and 1.75 °C immediately after the warm compress was removed. Arita et al. [37]. reported that the eyelid skin temperature increased by about 1.5 °C using the same device, and Purslow [39] reported that the upper and lower eyelid skin temperature increased, respectively, by 1.74 °C and 2.14 °C using the Blephasteam device (Thea Pharmaceuticals, Newcastle-under-Lyme, UK). In this study, the palpebral conjunctival temperature increased by 1.40 °C. Bilkhu et al. [40] reported that the palpebral conjunctival temperature increased by 1.3 °C using the MGDRx EyeBag (The EyeBag Company, West Yorkshire, UK). Some researchers [37,39] have reported that the corneal temperature increased immediately after warm compresses were removed by from 1 °C to 3 °C, and the current study observed an increase of 2 °C, which agreed with the previous results reported.

In experiment 1, the blood flow increased when the temperature increased. The response in blood flow to local heating has two stages [41,42,43]. First, the blood flow increased temporarily due to an axon reflex, i.e., the response elicited by sensory nerve stimulation and released calcitonin gene-related peptide (CGRP), followed by a slight decrease. Second, nitric oxide was secreted because CGRP stimulates vascular endothelial cells, resulting in continuously dilated blood vessels and increased blood flow until a plateau is reached. These responses were significant with rapid local heating than slow local heating [34,35]. Miyaji et al. reported that eyelid skin blood flow by LSFG did not increase significantly when the eyelid skin was warmed slowly using a stimulator (1 × 1 cm) that increased by 2 °C in a minute from 20 °C to 40 °C [44]. Gazerani and Arendt-Nielsen reported that the forehead blood flow increased significantly as a result of local rapid heating using a stimulator with a temperature of 43 °C for 1 min [45]. Wang et al. reported that the EyeBag heated in a microwave was significantly more effective in raising the temperature in the outer and inner eyelid skin temperature compared to the activated eye mask [46]. The current study suggested that blood flow increased because of use of a warming device (20 × 9 cm) heated in a microwave and a local rapid heating stimulation to the ocular anterior segment at about 51 °C.

In experiment 2, in contrast, when the blood flow increased as the result of oral capsaicin stimulation in the same manner as in a previous study [32], the temperature increased. Oral stimulation by capsaicin reportedly increased the blood flow in the entire face [32] as a result of the axon reflex [47], parasympathetic reflex vasodilation [48,49], and release of CGRP elicited by the sensory nerve [50,51]. Kashima and Hayashi reported that the blood flow in the entire face including the eyelid skin, forehead, cheek, nose, and upper and lower lip increased after oral capsaicin stimulation and the eyelid skin blood flow increased by about 20% [32]. In this study, the upper and lower eyelid skin blood flow increased by about 40% and 70%, respectively, using the same concentration of capsaicin. This may stem from methodologic differences. Kashima and Hayashi [32] evaluated the blood flow in the entire eyelid with the eyes closed, whereas we evaluated the upper and lower eyelid skin separately with the eyes open. Moreover, although they did not mention about bulbar conjunctival blood flow, we clarified that the blood flow increased not only in the entire face but also in the bulbar and palpebral conjunctiva using the same stimulation.

The pain scores resulting from oral stimulation were correlated with the blood flow and temperature in the upper and lower eyelid skin and the palpebral and bulbar conjunctiva. Moreover, a significant weak correlation was seen between the changes in the blood flow and the differences in temperature before and during stimulation at all locations. Chen et al. reported that the conjunctival blood flow but not the vessel diameter increased after wind stimuli applied to the corneal surface [21]. Nosch et al. reported that wind stimuli applied to the cornea at almost the same temperature as the cornea resulted in an increased corneal temperature when the stimulation was stronger [52]. Fricova et al. reported that inflammation associated with orofacial pain increased in temperature in the facial region of the reported pain [53]. The pain sensation was thought to be related closely to blood flow and/or temperature. In addition, we clarified for the first time that blood flow and temperature also are related in the ocular anterior segment in the present study. Bulbar conjunctiva had the greatest effect on the corneal temperature in the multiple regression analysis. Anterior segment temperature varied with not only blood flow but also ambient temperature [13] and tear film thickness [12]. Moreover, bulbar conjunctiva and cornea are adjacent tissues and affected similarly by tear film thickness. Therefore, there is a possibility that bulbar conjunctiva affected the corneal temperature most. Moreover, bulbar conjunctiva and cornea are adjacent tissues and are affected similarly by tear film thickness. Therefore, there is a possibility that bulbar conjunctiva affected the corneal temperature most.

The current study had some limitations. First, we did not directly instill capsaicin ophthalmic solution into the eye to stimulate the sensory nerve but used oral capsaicin stimulation because of ethical issues. Moreover, the ophthalmic solution could not be adjusted exactly to the corneal temperature. Second, LSFG measured the blood flow at a depth of about 1000 μm [31]. Akagi et al. [54] reported that the superficial layer extending from the bulbar conjunctival epithelium to a depth of 200 μm mainly has the conjunctival blood flow, and the deep layer from a depth of 200 to 1000 μm extending from the bulbar conjunctival epithelium mainly has the intrascleral blood flow. The blood flow at the bulbar conjunctival measured by LSFG may represent the total blood flow of both the bulbar conjunctiva and sclera. In the future studies, we will need to investigate the relationship between MBR measured by LSFG and FLSB method, which is currently widely used to assess conjunctival blood flow. Finally, the current study enrolled normal subjects. Some researchers have reported that the temperatures of the bulbar conjunctiva and eyelid skin in patients with dry eye were lower than those of normal subjects [6,15]. In the future, we will investigate the blood flow and temperature in patients with dry eye and contact lens wearers reporting discomfort.

## 5. Conclusions

In conclusion, we found that blood flow and temperature were correlated in the ocular anterior segment and that the temperatures of the ocular anterior segment and palpebral conjunctival blood flow are important factors contributing to the corneal temperature.

## Figures and Tables

**Figure 1 diagnostics-10-00695-f001:**
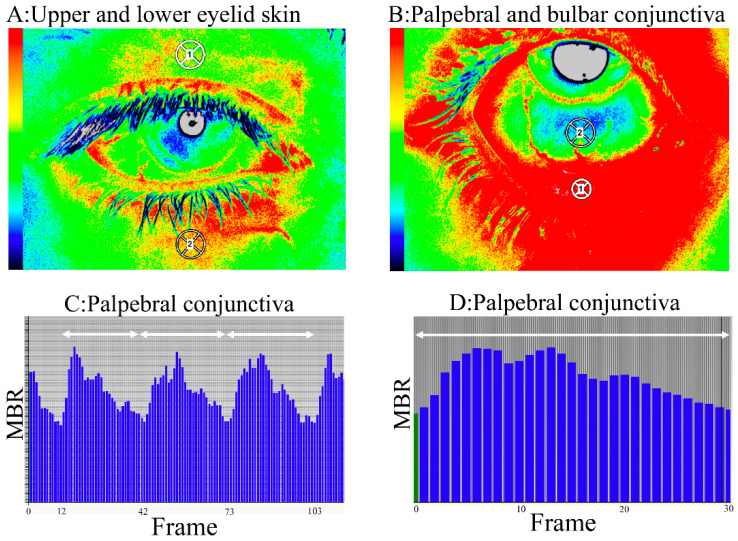
The method for calculating the mean blur rate (MBR) in the ocular anterior segment using laser speckle flowgraphy (LSFG). (**A**,**B**) Composite color map using the MBR. Red indicates high blood flow and blue indicates slow blood flow. The circle calculated is at the area of the upper (circle 1) and lower (circle 2) eyelid skin (**A**) and palpebral (circle 1) and bulbar (circle 2) conjunctiva (**B**). (**C**) The pulse waves show changes in the MBR, which is tuned to the cardiac cycle for 4 s. Each arrow indicates one pulse. (**D**) The total number of frames is 118. Normalization stabilizes two or more pulse waves into one pulse.

**Figure 2 diagnostics-10-00695-f002:**
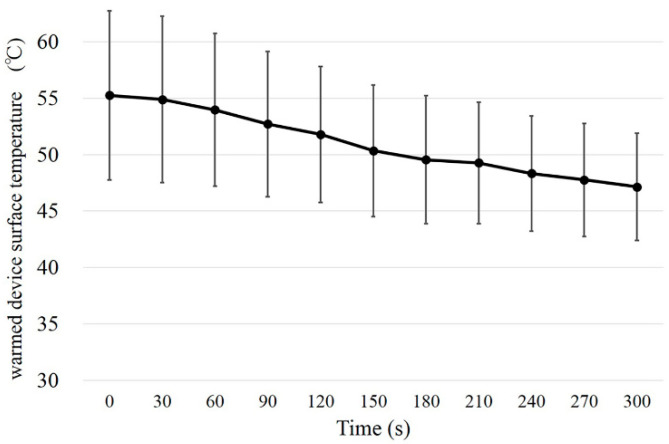
Results of device surface temperature profiles during 5 min after being microwaved for 30 s at 600 watts.

**Figure 3 diagnostics-10-00695-f003:**
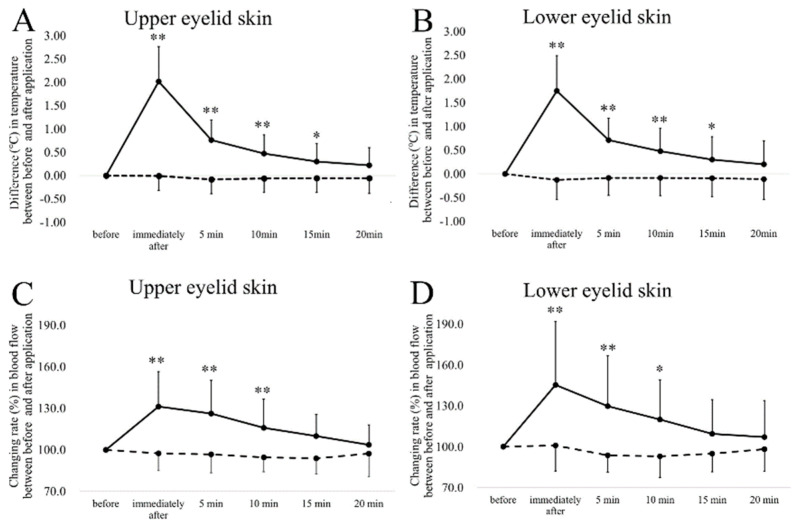
The time course of changes in the (**A**,**C**) upper and (**B**,**D**) lower eyelid skin temperature and blood flow after the application of warmed and room temperature devices for 5 min. The solid line indicates the warmed device; the dashed line indicates the room temperature control device. The error bars represent the standard deviations of the subjects. * *p* < 0.05 (Dunnett test), ** *p* < 0.01 vs. before device application.

**Figure 4 diagnostics-10-00695-f004:**
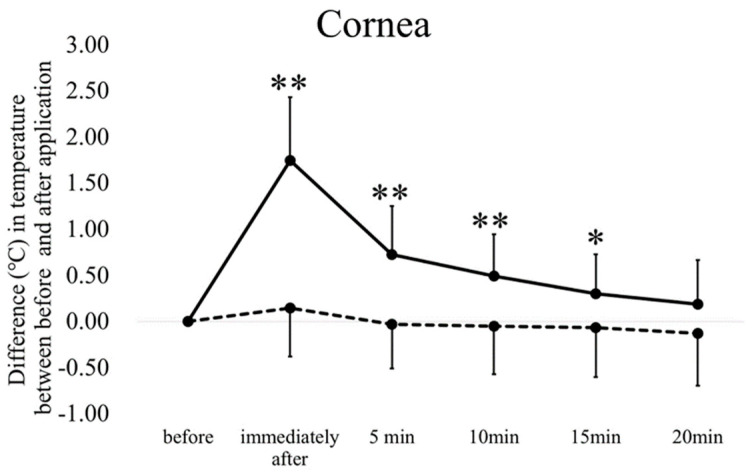
The time course of changes in the corneal temperature after the application of warmed and room temperature devices for 5 min. The solid line indicates the changes in temperature with the warmed device; the dashed line indicates the changes with the room temperature control device. The error bars represent the standard deviations of the subjects. * *p* < 0.05 (Dunnett’s test), ** *p* < 0.01 vs. before device application.

**Figure 5 diagnostics-10-00695-f005:**
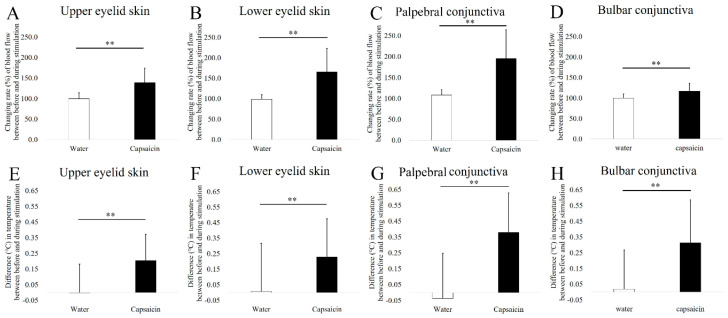
Comparison of (**A**,**E**) upper eyelid skin, (**B**,**F**) lower eyelid skin, (**C**,**G**) palpebral conjunctiva, and (**D**,**H**) bulbar conjunctiva temperature and blood flow between the water and capsaicin solutions. The asterisks indicate significant differences between solutions (** *p* < 0.01, paired *t*-test).

**Figure 6 diagnostics-10-00695-f006:**
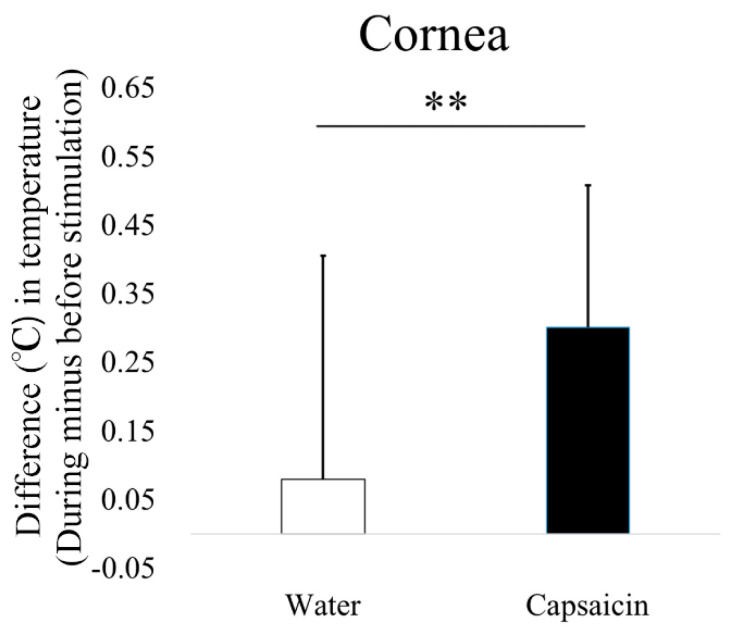
Comparison of corneal temperature in water and capsaicin solutions. The asterisks indicate significant differences between solutions (** *p* < 0.01, paired *t*-test).

**Figure 7 diagnostics-10-00695-f007:**
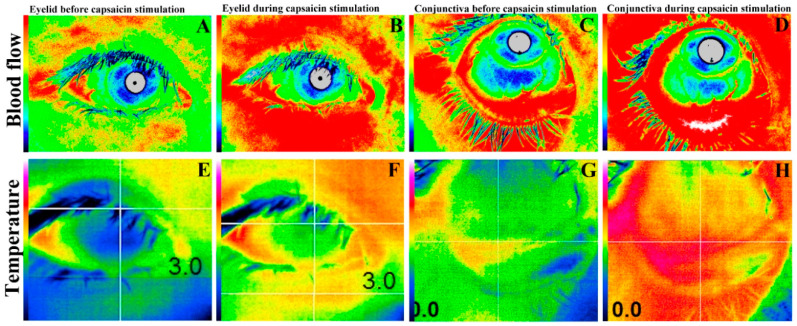
Representative blood flow and temperature images. Upper/lower eyelid skin blood flow (**B**) and temperature (**F**) during capsaicin stimulation show significant increase compared of those before stimulation (**A**,**E**). In the same way, palpebral and bulbar conjunctiva (**D**,**H**) and cornea (**F**) are significantly increased during stimulation compared of those before stimulation (**C**,**G**,**E**).

**Figure 8 diagnostics-10-00695-f008:**
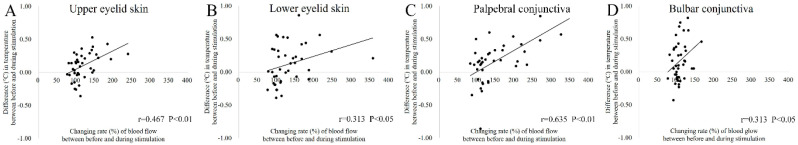
Correlation between the differences in temperature and changing rates of blood flow in the (**A**) upper eyelid skin (Pearson’s correlation coefficient, r = 0.467, *p* < 0.01); (**B**) lower eyelid skin (Pearson’s correlation coefficient, r = 0.313, *p* < 0.05); (**C**) palpebral (Pearson’s correlation coefficient, r = 0.635, *p* < 0.01) conjunctiva; and (**D**) bulbar conjunctiva (Pearson’s correlation coefficient, r = 0.313, *p* < 0.05).

**Table 1 diagnostics-10-00695-t001:** Result of Pearson’s coefficient of correlation between corneal temperature and ocular anterior-segment parameters.

Explanatory Variables	r Value	*p* Value
Upper eyelid skin temperature	0.619	<0.001
Lower eyelid skin temperature	0.573	<0.001
Palpebral conjunctiva temperature	0.781	<0.001
Bulbar conjunctiva temperature	0.867	<0.001
Upper eyelid skin blood flow	0.239	<0.05
Lower eyelid skin blood flow	0.081	0.470
Palpebral conjunctiva blood flow	0.228	<0.05
Bulbar conjunctiva blood flow	0.031	0.782

**Table 2 diagnostics-10-00695-t002:** Results of multiple regression analysis for independence of factors contributing to corneal temperature.

Variable			
Dependent	Independent	β	*p* Value
Corneal temperature	Upper eyelid skin temperature	0.291	<0.01
	Lower eyelid skin temperature	−0.369	<0.01
	Palpebral conjunctiva temperature	0.244	<0.05
	Bulbar conjunctiva temperature	0.762	<0.001
	Upper eyelid skin blood flow	0.054	0.410
	Palpebral conjunctiva blood flow	−0.159	<0.05

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
