# Peer review of "Correlation between Blood Flow and Temperature of the Ocular Anterior Segment in Normal Subjects"

_diagnostics, 2020, doi:10.3390/diagnostics10090695_

Round 1

Reviewer 1 Report

This publication reports on 2 experiments relating to blood flow and temperature changes of the anterior segment.  Increases were found with the application of warm compresses and capsaicin solution to the oral cavity and correlations were reported to the avascular corneal temperature.  This work has implications in understanding ocular physiology and also in disease pathology. 

Overall review comments

A relevant and interesting report of a clearly planned and executed study.

Abstract

Methods states that temperature was measured before and after application of warm compresses.  In the results statements are made ‘increased significantly until 15 minutes’, please clarify that this related to after the application of the warm compress.

Line 22: ‘Temperature of all place’? Perhaps ‘temperature of all locations’?

Introduction

An appropriate summary of the relevant research to this work that highlights the gap in current understanding and establishes the originality of the research. 

Methods

Well-constructed methodology with control of confounding factors and appropriate statistics applied to the data.

Figure 1 and 7 – color scales would be useful relating to the blood flow and temperature images.

Figure 1B:  It is not totally clear but the ‘tarsal’ measurement appears to be on the bulbar conjunctiva.  Tarsal should refer to the conjunctival on the inner side of the eyelid.

Page 4, line 161: Was there a reason why NIBUT was only recorded once?  Usually clinically it would be recorded 3 times and averaged.

Results

Graphs:  understand that Fig 3 might get too busy with gridlines but especially Figure 4 and 6 gridlines would be useful.

Figure 6: indicate that with the difference with capsaicin is an increase temperature, e.g. ‘During minus before stimulation’.

Discussion

A few questions from the results that were not addressed in the discussion, any thoughts on:

  • Why were the anterior segment temperatures more highly correlated to the corneal temperature but not so much blood flow measurements?
  • The multiple regression analysis had some negative values that were significant

Conclusions were consistent with the reported results.

Other

It may be a just a formatting issue but numbers (relating to the those listed in the text) were not visible next to the reference list.

Author Response

REVIEWER 1:

Comments:

Abstract

  1. [Line 17: Methods states that temperature was measured before and after application of warm compress. In the results statements are made ‘increased significantly until 15minutes’, please clarify that this related to after the application of the warm compress]

We appreciate the reviewer’s suggestion. As reviewer mentioned, Temperature and blood flow were increased significantly until 15 minutes after the application of the warm compress. We have added the sentence in the abstract (Lines 18).

  1. [Line 22: Temperature of all place? Perhaps ‘ temperature of all locations’? ]

We appreciate the reviewer’s comment. We have changed the sentence from ‘place’ to ‘locations’ (Lines 23).

Methods

  1. [Figure 1 and 7- color scales would be useful relating to the blood flow and temperature images]

We appreciate the reviewer’s suggestion. We have added the color scales at each image in Figure 1 and 7. Unfortunately, these color scales originally did not show the actual number of values (temperatures and/or MBR). So we have just added the color scales in each figure.

  1. [Figure 1B: It is not totally clear but the ‘tarsal’ measurement appears to be on the bulbar conjunctiva. Tarsal should refer to the conjunctival on the inner side of the eyelid]

We appreciate the reviewer’s suggestion. We changed the word from tarsal to palpebral in whole the manuscript. (Page 1. Lines 14, 18, 20, 22, 23. Page 3. Lines 96,121, 137. Page 4. Lines 148, 153, 158. Page 5. Lines 188, 190. Page 6. Lines 205, 211, 219. Page 7. Lines 224, 228, 233, Page 8. 241, 248, 253, 253, 256, 258, 260. Page 9. Lines 265, 266, 277, 283, 284, Page 10, Lines 316, 318. Figure1, Figure5, Figure 8, Table 1, Table 2).

  1. [Page 4, line 161: Was there a reason why NIBUT was only recorded once? Usually clinically it would be recorded 3 times and averaged]

We appreciate the reviewer’s comment. Due to the difference in measurement method, DR-1 interferometer BUT (NIBUT) records about 0-2 seconds longer than fluorescein BUT. For that reason, NIBUT measurement tend to occur reflex tearing. To avoid reflex tearing, some researchers measured once. We added the reason and references why only recorded once (Page 4. lines 163-164).

Results

  1. [Graphs: understand that Fig 3 might get to busy with gridlines but especially Figure 4 and 6 gridlines would be useful]

We appreciate the reviewer’s suggestion. We eliminated the figure of palpebral and bulbar conjunctival temperature and blood flow in figure 3. Moreover, we added the value of conjunctival temperature and blood flow when we recognized significant difference (page 5. Lines190-191, page 6. Lines 208-213).

  1. [Figure6: indicate that with the difference with capsaicin is an increase temperature, e.g. ‘During minus before stimulation’.]

We appreciate the reviewer’s suggestion. We changed the words. (Figure6).

Discussion

  1. [Why were the anterior segment temperatures more highly correlated to the corneal temperature but not so much blood flow measurements?]

We appreciate the reviewer’s suggestion. Anterior segment temperature also varied with not only blood flow but also ambient temperature and tear film thickness. Moreover, cornea and bulbar conjunctiva are adjacent tissues. Therefore, we thought that anterior segment temperature more highly correlated to the corneal temperature.

We have added the sentence (Page 10. Lines 327-332).

  1. [The multiple regression analysis had some negative values that were significant]

We appreciate the reviewer’s suggestion. We have changed the sentence (Page 10. Lines 212-125).

As reviewer know, standard partial regression coefficient (β) is an index to see the impact on the dependent factor, and the highest absolute value represent the higher effect on the dependent factor. In this multiple regression analysis, bulbar conjunctiva had the greatest effect on the corneal temperature. We have added the sentence (Page 10. Lines 327-332).

Other

  1. [It may be a just a formatting issue but numbers (relating to the those listed in the text) were not visible next to the reference list.]

We appreciate the reviewer’s comment. The word file shows it correctly, so it is formatting issue, but I will try to get it correct.

Reviewer 2 Report

Well conceived, interesting and relevant assessment of corneal temperature and its relationship to ocular tissue blood flow.  Using two separate methods of adjusting blood flow was very helpful.

I would suggest additional description of the use of the Warming Eye Pillow.  It was not clear to me whether this was used over the entire orbital area with the lids closed, or up against the eye with the lids open.  If used over the whole eye with lids closed, can we be certain the temperature of the anterior segment did not increase from direct heating?  Please support this assertion, assuming the anterior segment change was strictly dependent on blood flow change in the lids.

Author Response

REVIEWER 2:

  1. [ I would suggest additional description of the use of the warming eye pillow. It was not clear to me whether this was used over the entire orbital area with the lids closed, or up against the eye with the lids open. If used over the whole eye with lids closed, can we be certain the temperature of the anterior segment did not increase from direct heating? Please support this assertion, assuming the anterior segment change was strictly dependent on blood flow change in the lids.]

We appreciate the reviewer’s comment. The size of warming eye pillow was 20×9cm. we used over the entire orbital area with the eye opening in natural blinking. We added the sentence (Page 3. Lines 94)